# The Viral Protein Poly(A) Polymerase Catalytic Subunit Interacts with Guanylate-Binding Proteins 2 to Antagonize the Antiviral Ability of Targeting Ectromelia Virus

**DOI:** 10.3390/ijms242115750

**Published:** 2023-10-30

**Authors:** Zhenzhen Gao, Xiaobing He, Guohua Chen, Yongxiang Fang, Zejing Meng, Huihui Tian, Hui Zhang, Zhizhong Jing

**Affiliations:** 1State Key Laboratory for Animal Disease Control and Prevention, Ministry of Agriculture Key Laboratory of Veterinary Public Health, Lanzhou Veterinary Research Institute, Chinese Academy of Agricultural Sciences, Lanzhou 730046, China; m13321288521@163.com (Z.G.); hexiaobing@caas.cn (X.H.); chengguohua78@163.com (G.C.); yongxiangf@163.com (Y.F.); tianhuihui2208@163.com (H.T.); 17300223655@163.com (H.Z.); 2School of Public Health, Lanzhou University, Lanzhou 730000, China; mengzejing2022@163.com

**Keywords:** ectromelia virus (ECTV), poly(A) polymerase catalytic subunit (PAPL), guanylate-binding protein (GBP), ISG

## Abstract

The recent spread of the monkeypox virus among humans has heightened concerns regarding orthopoxvirus infections. Consequently, conducting a comprehensive study on the immunobiology of the monkeypox virus is imperative for the development of effective therapeutics. Ectromelia virus (ECTV) closely resembles the genetic and disease characteristics of monkeypox virus, making it a valuable research tool for studying orthopoxvirus–host interactions. Guanylate-binding proteins (GBPs), highly expressed interferon-stimulated genes (ISGs), have antagonistic effects against various intracellular pathogenic microorganisms. Our previous research has shown that GBP2 has a mild but statistically significant inhibitory effect on ECTV infection. The presence of a significant number of molecules in the poxvirus genome that encode the host immune response raises questions about whether it also includes proteins that counteract the antiviral activity of GBP2. Using IP/MS and co-IP technology, we discovered that the poly(A) polymerase catalytic subunit (PAPL) protein of ECTV is a viral regulatory molecule that interacts with GBP2. Further studies have shown that PAPL antagonizes the antiviral activity of GBP2 by reducing its protein levels. Knocking out the *PAPL* gene of ECTV with the CRISPR/Cas9 system significantly diminishes the replication ability of the virus, indicating the indispensable role of PAPL in the replication process of ECTV. In conclusion, our study presents preliminary evidence supporting the significance of PAPL as a virulence factor that can interact with GBP2.

## 1. Introduction

Orthopoxviruses have played a significant role in human history, particularly the variola virus, which is responsible for smallpox. Smallpox has been one of the deadliest human pathogens, causing an estimated 300–500 million deaths worldwide. In fact, it has caused more human deaths than all other infectious diseases combined throughout history [1]. Thanks to extensive research and successful vaccination efforts, smallpox was ultimately eradicated, representing a remarkable accomplishment in the field of immunology [2,3,4]. Despite its eradication, there is still concern about orthopoxvirus infections, as the variola virus could potentially be used as a bioterror weapon [5,6]. Since the first human infection with monkeypox virus was reported in the UK in 2022, it has become endemic in over 100 countries and continues to spread from person to person. The virus is also evolving to better target the human immune system [7]. The monkeypox virus has the ability to infect not only monkeys and humans, but also rodents and primates, which serve as natural reservoirs and hosts for the virus. Conducting live virus experimental activities poses significant biosafety risks, thereby restricting the study of infection and immune mechanisms [8]. Ectromelia virus (ECTV) belongs to the *Orthopoxvirus* genus of the *Chordopoxviridae* subfamily. It is a large, linear, double-stranded DNA (dsDNA) virus with an envelope. In experimental conditions, ECTV can be transmitted between wild and laboratory mice. Acutely infected mice also develop a pustular rash and experience fulminant hepatitis, resembling smallpox in mice [9,10]. The virus model has become an ideal alternative for studying the biological characteristics of orthopoxviruses. Poxviruses have large genomes, ranging from 130 to 365 kb, and encode approximately 200 genes. During their co-evolution with natural hosts, poxviruses have developed complex host-specific strategies to efficiently replicate by controlling the antiviral responses [11,12,13]. Roughly half of the encoded proteins interact with antiviral or anti-inflammatory pathways, such as interferon (IFN), apoptotic factors, and inflammatory factors [14]. The presence of numerous immunomodulatory genes in poxviruses could be attributed to the large virions, which limits their ability to rapidly mutate and evade adaptive immune responses, necessitating the employment of more intricate strategies to counteract the immune system. 

Several reports have indicated that IFN-α/β and IFN-γ are crucial in the recovery process from ECTV infection [15]. However, interferon does not directly act on pathogens itself. Instead, it acts indirectly through autocrine and paracrine pathways. This occurs after binding to the specific receptor IFNAR1/IFNAR2 located on the surface of target cells, which then initiates signals through the classic JAK-STAT pathway. Consequently, a large number of ISGs are transcribed, providing protection against viral infection. The earliest identified ISGs, such as Mx1, PKR, and OAS1, exhibit potent antiviral activity [16]. As new antiviral ISGs are discovered, relatively few of them have been fully characterized for their role in virus infection. GBPs are strongly induced ISGs and contribute to cell-autonomous immunity. Previous research has demonstrated the crucial role of GBPs in host defense against viral, bacterial, and protozoan pathogens [17]. Apart from directly exerting antiviral functions against certain viruses, some GBPs also function as regulators in host antiviral innate immunity, particularly against RNA viruses [18]. However, current research on the role of GBP in DNA viruses is still quite limited. GBPs belong to the dynamin GTPase superfamily, characterized by their high GTPase activity, which is further enhanced by oligomerization and interaction with endomembrane. Activation of GTPases on the membrane leads to changes in membrane conformation, resulting in division or fusion, which regulates cell growth and resistance to pathogen infection [19]. Our previous study revealed a significant upregulation of GBPs expression upon ECTV infection [20]. Furthermore, we discovered that GBP2 exhibited a mild but statistically significant inhibitory effect on ECTV infection. GBP2 primarily functions through its N-terminal large GTPase (LG) domain. The antiviral activity of GBP2 against ECTV is completely lost in the GBP2-ΔLG mutant. However, the GBP2-ΔCTHD mutant, which has a deletion in the C-terminal α-helix, retains its antiviral activity. Furthermore, Lys-51 of GBP2 is identified as the key site for ECTV inhibition [21]. There have been limited reports on the activity of poxvirus-targeted ISGs. Notable examples include the VACV E3 and K3 proteins, which can prevent dsRNA-induced PRR activation. When cells are infected with ECTVΔE3L, viral replication is halted [22]. However, the ECTV gene encoding K3L is disrupted due to the premature appearance of a stop codon [23]. The EVM002 protein encoded by ECTV has the ability to inactivate the NF-κB subunit p50 [24], which leads to a reduction in the expression of pro-inflammatory cytokines in the body [25]. 

In order to investigate whether ECTV encodes proteins that counteract the host’s GBP immune response, we conducted immunoprecipitation on RAW264.7 cells infected with ECTV using mouse GBP1-5 monoclonal antibodies. Subsequently, mass spectrometry analysis was performed on the precipitates to identify viral proteins that potentially interact with GBP family molecules. The findings from immunoprecipitation and Western blotting analysis revealed an interaction between the viral poly(A) polymerase catalytic subunit (PAPL) protein of ECTV and GBP2. Understanding the mutual regulatory mechanism between viral proteins and host immune responses is crucial for comprehending viral pathogenesis and developing new treatments. Further investigation of these genes could provide valuable insights into novel host antiviral mechanisms and shed light on how viruses exploit host proteins to evade the immune system. Additionally, they contribute significantly to the advancement of antiviral drugs and vaccines.

## 2. Results

### 2.1. Identification of ECTV Proteins That May Interact with GBP2

Poxviruses have developed sophisticated strategies to evade the immune system and produce a wide range of proteins that can manipulate host innate and adaptive immune responses of [26]. To investigate whether ECTV encodes proteins that interact with GBP2, RAW264.7 cells were infected with ECTV and subjected to immunoprecipitation/mass spectrometry analysis to identify potential protein interactions with GBPs. The analysis revealed the precipitation of six viral proteins: poly(A) polymerase large subunit (PAPL), glutaredoxin-1, intermediate transcription factor 3 large subunit (VIT3L), semaphorin-like protein (SEMA), profilin, Kelch repeat, and BTB domain-containing protein 1(KBTB1). These proteins are encoded by the ECTV genes *PAPL* (ORF, EVM041), *EVM053*, *VIT3L* (ORF, EVM126), *EVM139*, *EVM141*, and *KBTB1* (ORF, EVM150), respectively (Table 1).

The antibody used in this study, anti-GBP1-5, has the ability to react with GBP1, GBP2, GBP3, GBP4, and GBP5. In order to investigate the interaction between GBP2 and the six viral proteins identified through mass spectrometry analysis, we generated recombinant plasmids of viral proteins tagged with HA and co-transfected them with Flag-GBP2 into HEK-293T cells. Subsequently, co-immunoprecipitation (co-IP) experiments were conducted using anti-Flag monoclonal antibodies. The findings revealed that three of these proteins, including HA-PAPL, HA-glutaredoxin-1, and HA-profilin, exhibited a preliminary co-precipitation reaction with GBP2 (Figure 1a). To investigate the impact of these three precipitated viral proteins on GBP2 expression, we conducted co-transfection experiments in HEK-293T cells. Equal amounts of HA-PAPL, HA-glutaredoxin-1, or HA-profilin viral proteins were co-transfected with Flag-GBP2. Subsequently, Western blotting analysis was performed. The results showed that HA-PAPL significantly reduced the protein level of Flag-GBP2 (Figure 1b). Moreover, increasing the amount of HA-PAPL resulted in a dose-dependent reduction in Flag-GBP2 expression (Figure 1c). Based on these findings, it can be inferred that HA-PAPL can interact with Flag-GBP2 and reduces its expression in a dose-dependent manner. However, the effects of HA-glutaredoxin-1 and HA-profilin on Flag-GBP2 were not clearly observed. As our focus is on exploring the viral proteins that regulate GBP2 immune responses, our upcoming experiments will concentrate on confirming the interaction between PAPL and GBP2.

### 2.2. The Viral Protein PAPL of ECTV Interacts with GBP2

To further confirm the interaction between PAPL and GBP2, we conducted reverse co-IP verification using HA antibody. The interaction between PAPL and GBP2 was also detected through endogenous co-IP, GST-pulldown, and IFA experiments. In the first step, we co-transfected Flag-GBP2 and HA-PAPL recombinant plasmids into HEK-293T cells, while Flag-GBP2 and HA empty were co-transfected as a negative control. Protein samples were collected and verified by co-IP with anti-HA antibody. Co-IP was also performed with normal IgG as a negative control. The results demonstrated that only HA-PAPL was able to precipitate Flag-GBP2 when using anti-HA antibody for co-IP (Figure 2a). Additionally, we treated RAW264.7 cells with IFN-γ or infected them with ECTV to investigate the interaction between endogenous GBP2 and PAPL. The results revealed that GBP2 interacts with PAPL produced by ECTV (Figure 2b). GST-pulldown was performed to investigate the direct binding between PAPL and GBP2 in vitro. GST, which stands for glutathione-S-transferase protein, was used to stably combine with glutathione. Glutathione–Sepharose 4B beads were used to bind the GST tagged protein and act as a bait protein. When a target protein interacts directly with the bait protein in the system, it can be pulled down. Our results demonstrated that GST-GBP2 successfully pulled down a significant amount of His-PAPL, while GST had no noticeable pull-down effect (Figure 2c). These findings indicate a direct interaction between PAPL and GBP2. Fluorescence colocalization analysis refers to the examination of the overlap between multiple fluorescent markers of varying colors within the same spatial region. Our confocal microscopy assay showed that HA-PAPL and Flag-GBP2 were both located in the cytoplasm and exhibited a high degree of colocalization, with a coefficient of 0.933 (Figure 2d). Taken together, the above results indicate that GBP2 can interact with viral protein PAPL.

### 2.3. Identification of Key Sites for Interaction between PAPL and GBP2

The molecular weight of GBP family proteins is approximately 65–67 kDa and consists of three main domains: the N-terminal large GTPase (LG) domain, the middle domain (MD) connected to the LG domain through the hinge region, and the C-terminal GTPase-directing domain (GED) [27]. The MD and GED domains together are referred to as the C-terminal α-helical domain (CTHD). To identify the key domains and sites for the interaction between PAPL and GBP2, we utilized overexpression plasmids of GBP2 truncation mutants (Flag-GBP2-ΔLG, Flag-GBP2-ΔCTHD, Flag-GBP2*^R48A^*, Flag-GBP2*^K51A^*) that were previously constructed (Figure 3a,b). Co-IP was performed to analyze the interaction between these plasmids and HA-PAPL, while an empty EV was used as a negative control. The findings demonstrated that both full-length Flag-GBP2 and Flag-GBP2-ΔCTHD were able to interact with HA-PAPL, whereas Flag-GBP2-ΔLG was unable to do so (Figure 3c). This suggests that the LG domain is crucial for facilitating the interaction between GBP2 and PAPL. Additionally, we found that Flag-GBP2*^R48A^* co-precipitated with HA-PAPL, whilst Flag-GBP2*^K51A^* did not (Figure 3d). The confocal imaging indicates that the colocalization coefficient between Flag-GBP2*^R48A^* and HA-PAPL was significantly higher than that of Flag-GBP2*^K51A^*. The coefficient of the former was 0.902, while that of the latter was only 0.761 (Figure 3e). These results show that the K51 residue of GBP2 plays a critical role in its interaction with PAPL.

PAPL, a polymerase present in orthopoxviruses, is responsible for elongating the 3′-end of DNA or RNA in a template-independent manner, ultimately leading to the formation of a 3′-poly(A) tail [28]. This polymerase contains two ATP-binding and activation sites, specifically located at aspartic acid residues 202 and 204 [29]. To investigate whether the mutation of PAPL affects its interplay with GBP2, we generated two mutant PAPL plasmids (PAPL*^D202G^* and PAPL*^D204G^*) with an HA tag. The results showed that both HA-PAPL*^D202G^* and HA-PAPL*^D204G^* could bind Flag-GBP2 and Flag-GBP2-ΔCTHD (Figure 3f), suggesting that the mutation of PAPL had little effect on its interaction with GBP2. 

### 2.4. PAPL Antagonizes the Antiviral Activity of GBP2

Our experiments have demonstrated an interaction between PAPL and GBP2, revealing that HA-PAPL can effectively reduce the expression of Flag-GBP2 in a dose-dependent manner. However, the impact of PAPL on the antiviral properties of GBP2 is still uncertain. In order to investigate the influence of PAPL on GBP2’s ability to inhibit ECTV replication, as well as the impact of PAPL alone on viral replication, we conducted experiments where HA-PAPL was overexpressed either individually or in conjunction with Flag-GBP2. Our findings indicate that the overexpression of HA-PAPL alone does not have a significant direct impact on ECTV viral replication. However, when HA-PAPL and Flag-GBP2 are co-expressed, the inhibitory effect of Flag-GBP2 on ECTV gradually diminished as the amount of HA-PAPL increased. Increasing the dose of HA-PAPL resulted in a gradual decrease in the protein level of Flag-GBP2, while the virus titers and viral protein level showed a gradual recovery (Figure 4a,b). Additionally, we investigated the impact of HA-PAPL*^D202G^* and HA-PAPL*^D204G^* mutations on the antiviral activity of Flag-GBP2. These mutations targeted two crucial active sites of PAPL. Our findings revealed that both mutant plasmids resulted in a weakened ability to inhibit the expression of Flag-GBP2 protein. Moreover, these mutations also diminished the antagonistic effect of Flag-GBP2 against ECTV (Figure 4c,d). These findings indicate that although HA-PAPL itself does not significantly promote viral replication, it does weaken the antiviral properties of Flag-GBP2. 

### 2.5. Deletion of PAPL Gene of ECTV Reduces Virus Replication

In order to further investigate the role of PAPL in viral replication and its regulatory effect on GBP2, we employed a CRISPR-Cas9 system to delete the *PAPL* gene to construct a PAPL knockout ECTV. The recombinant ECTV expressing EGFP (ECTV-EGFP), which was previously constructed in our laboratory, was used as the parent virus. The gene editing vector Genloci pGK1.1, which is based on the new generation of artificial nucleic acid CRISPR/Cas9, is utilized in this system [30]. The expression of the gRNA sequence is guided by the U6 promoter. Figure 5a illustrates the vector construction designed for targeted knockout of the *PAPL* gene using the CRISPR-Cas9 system (Figure 5a). The 20 bp base sequence depicted in the figure represents the reverse start guide oligo DNA sequence located at the 600 bp base of the *PAPL* gene. The term ‘BGHpA’ on the right refers to the Bovine Growth Hormone (BGH) polyadenylation (polyA) signal sequence. The poly(A) tail serves to protect the mRNA from exonuclease attack. BSR-T7 cells were transfected with the plasmid containing a gRNA sequence specifically targeting PAPL and subsequently infected with ECTV-EGFP. Following about 10 rounds screening, a PAPL deletion strain with a frameshift mutation near the key active site of PAPL (582 bp) was successfully obtained. The proteins of the screening virus, as well as the wild-type ECTV (ECTV-WT) and ECTV-EGFP parent viruses, were collected by infecting BSR-T7 cells. Detection was performed using viral H3L antiserum and PAPL antiserum. Western blotting results showed that the PAPL protein could not be detected in the screening virus, and the expression of H3L protein was also reduced. Yet, ECTV-WT and ECTV-EGFP parent viruses exhibited high expression of H3L and PAPL proteins (Figure 5b). This indicates that we successfully screened the PAPL knockout ECTV, and it was named ECTV-EGFP-ΔPAPL. The viral plaque assay was conducted to evaluate the effect of this deletion on viral replication. The results showed that ECTV-EGFP-ΔPAPL had significantly reduced replication ability compared to the parent virus. Additionally, the plaques formed by ECTV-EGFP-ΔPAPL were smaller in size compared to the parent ECTV-EGFP and ECTV-WT (Figure 5c). Furthermore, we examined the in vitro growth characteristics of ECTV-EGFP-ΔPAPL in CV-1 cells and compared it with ECTV-WT and ECTV-EGFP using a multistep growth curve analysis. The findings revealed that ECTV-EGFP-ΔPAPL exhibited significantly delayed growth kinetics compared to ECTV-WT or parent ECTV-EGFP (Figure 5d).

Mouse macrophages are the primary cells responsible for producing GBPs and initiating the inflammatory immune response upon ECTV infection. These cells are well-suited for studying the natural immune response to ECTV. In our previous study, we have screened GBP2 knockdown RAW264.7 cell lines (sh-GBP2) using the GBP2 sh-RNA lentiviral particles. qPCR analysis revealed a significant reduction in *GBP2* expression in the sh-GBP2 cells compared to the control cell lines (sh-Con). Additionally, Western blotting analysis confirmed a significant decrease in GBP2 expression in the sh-GBP2 cells following ECTV infection. These findings indicate the successful screening of the GBP2 knockdown cell line [21]. To investigate the impact of knocking out the *PAPL* gene on the antiviral activity of GBP2, we infected sh-GBP2 and sh-Con cells with ECTV-EGFP or ECTV-EGFP-ΔPAPL. After 24 h of infection, we collected cell samples to analyze the expression of proteins. Additionally, we collected samples at 6 h, 24 h, and 48 h post-infection (hpi) to extract the viral genome and measure changes in the viral DNA copy number. The findings revealed a significant increase in viral genomic copies and viral protein level when GBP2 was knocked down during ECTV-EGFP infection. There was no significant difference in viral genomic copies at 6 hpi (Figure 5e,f). This confirms our previous findings that GBP2 plays a crucial role in limiting ECTV replication, particularly in the late stage of viral replication. On the other hand, when cells were infected with ECTV-EGFP-ΔPAPL, there was no significant difference in viral replication levels between the GBP2 knockdown and control groups. Only at 48 hpi did knocking down GBP2 slightly promote the increase in viral DNA copy numbers (Figure 5f). Simultaneously, deletion of the *PAPL* gene resulted in a significant decrease in viral replication ability, with the viral H3L protein level and viral DNA copy number being significantly lower than that of ECTV-EGFP (Figure 5e,f). The results indicate that PAPL plays a significant role in ECTV replication, and its deletion greatly reduces virus replication. However, knocking down GBP2 does not rescue the damage to viral replication. These findings suggest that PAPL is an important candidate virulence factor of ECTV. 

## 3. Discussion

Poxviruses serve as valuable models for investigating genome evolution. They demonstrate relatively low rates of point mutation accumulation [31], but frequently undergo gene duplication, loss, and gain through horizontal gene transfer (HGT) and recombination between different species [23]. These events play a crucial role in host adaptation and evasion of host antiviral responses. The ability of a virus to replicate within cells hinges upon its effective regulation of the cell’s antiviral response [32]. Poxviruses impact the outcome of infection by encoding numerous viral factors that influence host range, virulence, and evasion of host innate and adaptive immune defenses. 

Through co-IP/mass spectrometry, we have identified a preliminary interaction between GBP2 and PAPL, glutaredoxin-1, and profilin. PAPL belongs to the poxviridae poly(A) polymerase catalytic subunit family, which is responsible for creating the 3′-poly(A) tail of mRNA. These poly(A) tails have long been recognized as stable 3′ modifications of eukaryotic mRNAs and are added during nuclear pre-mRNA processing [33]. Glutaredoxin-1 exhibits thioltransferase and dehydroascorbate reductase activities, utilizing glutathione as a cofactor to reduce disulfides in both prokaryotes and eukaryotes [34]. The vaccinia glutaredoxin O2L has been identified as a cofactor for viral ribonucleotide reductase [35,36]. Additionally, the *G4L* gene of vaccinia virus encodes another glutaredoxin that plays an essential role in the morphogenesis of vaccinia virus particles [37]. All orthopoxviruses sequenced so far encode profilin-like proteins, which share more than 90% amino acid identity. The profilin orthologues encoded by EVM141 have the ability to directly bind to cellular α-tropomyosin, facilitating the intracellular transport of viral proteins or intercellular spread of the virus particles [38]. However, the interactions between glutaredoxin-1, profilin, and GBP family members, as well as the underlying regulatory mechanisms, require further investigation.

Our study confirmed that PAPL can interact with GBP2 and strongly reduce its expression (Figure 1a–c). Further research is needed to determine if PAPL can also reduce the expression of other ISGs or even viral proteins. In co-IP experiments, the dosage ratio of PAPL to GBP2 and its truncation mutants is generally 1:1.5 due to the significant inhibitory effect. Therefore, the inhibition of GBP2 by PAPL may not be visually intuitive in the co-IP experiment. In Figure 1a, the expression levels of various viral proteins in cells exhibit significant variation. To observe the expression of all proteins simultaneously, the protein amount needs to be adjusted multiple times. It is important to note that in endogenous co-IP experiments, the expression of viral proteins did not change significantly after infection with ECTV, regardless of whether it was pre-treated with IFN-γ or not (Figure 2b). Poxviruses have evolved various strategies to evade the host’s interferon immune response. All *Orthopoxviruses* express IFN-binding proteins that effectively block IFN-mediated signaling cascades, thus inhibiting the activation of antiviral defense mechanisms [39]. ECTV vIFN-γR receptor has only 20% sequence similarity with the host receptor; it can still bind IFN-γ and inhibit its biological activity in vitro [40]. 

The most notable characteristic of poxvirus DNA replication is that it occurs in the cytoplasm rather than the nucleus. Unlike many other large DNA viruses, poxvirus genomes are replicated in viral factories located near the nucleus [41]. The GBP family is primarily found in the cytoplasm and can target different organelles and vesicles. In mouse macrophages and fibroblasts treated with IFN-γ, mGBP2 exhibited a granular distribution in the cytoplasm and was specifically associated with the perinuclear membrane compartment [42]. This close proximity may enhance the ability of GBP2 to target poxviruses. Confocal microscopy observation of the subcellular localization of GBP2 and PAPL revealed that Flag-GBP2 and HA-PAPL appeared to co-localize in the perinuclear region (Figure 2d and Figure 3c). Previous studies have indicated that GBPs primarily act on pathogenic microorganisms by binding to host membranes, pathogen-containing vacuole membranes, or directly targeting pathogens [43]. Additionally, it has been reported that MxA, another member of the GTPase superfamily, is recruited to the virus factory of ASFV to inhibit viral replication by suppressing the expression of late genes [44]. Further investigation is required to determine whether GBP2 is recruited to virus factories to exert antiviral effects.

Asp202 and Asp204 are conserved active sites across all orthopoxvirus poly(A) polymerase (PAP) families [29]. However, the results indicate that these mutations do not affect the interaction with GBP2 (Figure 3). Nonetheless, the mutated HA-PAPL loses its ability to inhibit ectopically expressed Flag-GBP2 and disrupts the antagonistic effect on GBP2’s antiviral activity. This suggests that the activity of PAPL binding to ATP analogs plays a crucial role in antagonizing GBP2, although the specific mechanism requires further investigation. In addition to its function as a Poly(A) polymerase, PAPL also weakens the host GBP immune response. However, increasing the dose of HA-PAPL alone does not significantly impact ECTV replication. Studies on VACV have shown that PAPL rapidly adds a 24–30 nucleotide poly(A) tail before dissociation of the RNA substrate, and elongation of the poly(A) tail is not necessary for vaccinia virus growth in cell culture. The viral poly(A) polymerase-stimulating function in cell culture appears to be an accessory function and not necessary for normal mRNA metabolism and function [28]. It is hypothesized that poxvirus is capable of expressing the poly(A) polymerase necessary for its own replication while also effectively controlling the host GBP immune response. This allows for a balance between virus amplification and the host’s immune response. The sole ectopic expression of HA-PAPL may enhance the activity of poly(A) polymerase and extend the poly(A) tail, but it has little effect on ECTV replication. Previous analysis of GBP*^chr3^* transcript levels showed that GBP*^chr3^* is strongly upregulated at 12 h after ECTV infection and then begins to decrease [21]. This suggests that ECTV may encode a protein that antagonizes GBP expression, or there may be other immune responses that inhibit GBP translation or rapidly degrade the GBP protein to ensure normal virus replication. However, the overexpression of Flag-GBP2 disrupts the balance in the culture medium, and the ectopic expression of HA-PAPL rapidly diminishes the expression of Flag-GBP2, thereby facilitating virus replication. The experiment used transient transfection, and a large amount of HA-PAPL expression plasmid was not used due to its toxicity to the cells. In future studies, a stable PAPL overexpression plasmid can be established to further explore the impact of PAPL on ECTV replication.

According to our previous data, ECTV infection resulted in a significant increase in the transcription of GBP*^chr3^*, particularly GBP2 [21]. Our results in this study showed that the protein levels of GBP2 were upregulated in sh-Con cells, with a higher upregulation level observed after infection with ECTV-EGFP-ΔPAPL compared to ECTV-EGFP infection. This suggests that PAPL plays a role in regulating the expression of GBP2, and deletion of *PAPL* gene increased the expression of endogenous GBP2 (Figure 5e). Furthermore, we observed that infection with either virus did not induce the upregulation of GBP2 expression in the sh-GBP2 cells, providing additional evidence of the effectiveness of knockdown. Further research is needed to determine whether the substantial decrease in ECTV-EGFP-ΔPAPL replication capacity is caused by the upregulation of GBP2 or other GBPs. These results imply that, under normal infection conditions, it is important to maintain a balanced expression of endogenous PAPL and GBP2 in macrophages. Knocking down GBP2 can enhance viral replication. However, the robust decrease in viral replication caused by PAPL knockout cannot be rescued by silencing GBP2. This suggests that PAPL not only regulates the expression of GBP2 but may also have an influence on the expression of other antiviral molecules. Further investigation is required to determine whether PAPL deletion impairs the formation of the viral poly(A) tail, leading to mRNA instability and subsequently reducing ECTV replication ability. Overall, these findings highlight the crucial role of PAPL in viral replication. Additional research is required to elucidate the precise mechanism of PAPL in ECTV replication and to assess the implications of PAPL knockout on the virus’s pathogenicity in animals.

This study provides valuable insights into the innovative strategies employed by poxviruses to regulate host responses.

## 4. Materials and Methods

### 4.1. Cells

CV-1 cells (African green monkey kidney fibroblast cells) were cultured in RPMI 1640 basic medium (C11875500BT, Gibco, Billings, MT, USA) containing 10% fetal bovine serum (FBS, 10091148, Gibco). BSR-T7 (derivative of baby hamster kidney cell), RAW264.7 (mouse macrophage), and HEK-293T (human embryonic kidney cells) were grown in Dulbecco’s Modified Eagle Medium (C11995500CP, DMEM; Gibco) containing 10% FBS. The media was supplemented with 1% penicillin–streptomycin (15140122, Gibco). These cells were preserved and passaged in our laboratory. CV-1 cells were utilized for virus propagation and viral titers detection. BSR-T7 cells have been genetically modified, resulting in significantly enhanced transfection efficiency (90%). These cells are capable of transiently transfecting plasmids using Lipofectamine 2000 and can be infected with viruses. RAW264.7 cells were utilized for preparing GBP2 knockdown cell lines and subsequent viral infection. HEK-293T cells were used for co-IP and plasmid co-transfection experiments via calcium chloride transfection method and are not susceptible to viral infection.

### 4.2. Virus

The viruses used in this study included ECTV-Moscow (EVM, ATCC 1374). ECTV expressing enhanced green fluorescent protein (ECTV-EGFP) was created through standard homologous recombination techniques [45,46]. The *EGFP* gene was inserted into a non-coding region of ECTV-Moscow, ensuring that the growth characteristics and infectivity of the virus were not affected. ECTV-EGFP-ΔPAPL was generated using a simple and efficient CRISPR-Cas9 system [30]. Briefly, the oligo DNA sequences targeting PAPL were designed using the Addgene online tool (http://crispor.tefor.net/) (accessed on 18 December, 2022). The two single-stranded oligo DNAs of each pair were denatured and then annealed to form dsDNA. This dsDNA was subsequently cloned into the pGK1.1 (Puro) vector. Plasmids were transfected into BSR-T7 cells and infected with ECTV-EGFP virus (MOI = 0.1). Samples were collected 48 h later. CV-1 cells were plated in a 6-well plate and infected with the viral lysate supernatant. Single clones were picked and subjected to PCR verification. This process was repeated for approximately ten rounds of passage, and the mutant virus strain of PAPL was obtained. Positive clones were screened using PCR and confirmed through nucleotide sequencing. All viruses were propagated in CV-1 cells and stored at −80 °C. The sequences of these gRNA oligos and the primers can be found in the Appendix A. The comparison result between the mutant sequence and the parental sequence can be found in Appendix A. 

### 4.3. Plaque Assay and TCID_50_

All viruses were titrated using CV-1 cells. The virus was released from infected cells through three cycles of freezing and thawing. For the plaque assay, the supernatant was serially diluted 10-fold in serum-free medium and 200 μL of the diluted solution was added to CV-1 cells in 24-well plates. The cells were incubated for two hours, after which the inoculums were replaced with overlay media containing 0.75% carboxy methyl cellulose and incubated at 37 °C for 5–7 days. Subsequently, the cells were fixed using 4% paraformaldehyde. They were then stained with 250 μL of a 1% crystal violet solution containing 20% ethanol for plaque counting. In the TCID_50_ assay, CV-1 cells were infected with the indicated viruses at a multiplicity of infection (MOI) of 0.1, and samples were collected at 2, 4, 8, 12, 18, 24, and 48 h post-infection. The viral lysates were serially diluted 10-fold in standard serum-free medium, and 100 μL of the diluted solution was added to CV-1 cells seeded in 96-well plates. Eight wells were infected for each dilution and further incubated for 5–7 days. Virus titers were calculated using the Reed–Muench method and expressed as Log10 (TCID_50_ /mL).

### 4.4. Antibodies

Primary antibodies: GBP1-5 (G-12) (sc-166960, Santa Cruz, Dallas, Texas, USA); GBP2 (11854-1-AP, Proteintech, Wuhan, China); Flag (F3165, Sigma-Aldrich, St. Louis, MO, USA); HA (ab9110, Abcam, Cambridge, UK); beta Tubulin (ab6046, Abcam); Myc (AB001-01, ACE, Changzhou, China); GAPDH (G8795, Sigma-Aldrich). Secondary antibodies: HRP-conjugated goat anti-mouse IgG (SA00001-1, Proteintech); HRP-conjugated goat anti-rabbit IgG (SA00001-2, Proteintech); Mouse Anti-Rabbit IgG LCS (A25022, Abbkine, Wuhan, China). Polyclonal EVM H3L and Polyclonal PAPL antiserum were produced in our laboratory. In brief, the H3L and PAPL target sequences were amplified by PCR using the genomic DNA of ECTV-Moscow to construct pET30a-H3L and pET30a-PAPL prokaryotic expression plasmids. These plasmids were then transformed into Rosetta competent cells. Positive clones were selected, and the culture was induced with IPTG (1 mmol/L). The induced cell sediments were dispersed by sonication then centrifuged to collect the precipitate. The precipitate was washed 3 times with 2M urea, resuspended in 8M urea, and placed in a shaker at 4 °C overnight for lysis. The supernatant was purified with a nickel chromatography column and eluted with imidazole. The protein was then subjected to gradient dialysis. The recombinant proteins fused with His tag and can be used in subsequent GST-pulldown experiments. To generate polyclonal antibodies, the renatured protein was immunized into rabbits at a dosage of 2.5 μg/rabbit. The antibody titers were detected, and the serum was collected after two immunizations.

### 4.5. Plasmids Construction and Transient Transfection

The full-length and truncated mutant mouse GBP2 Flag-tagged plasmids were previously constructed in our laboratory [21]. The *PAPL*, *EVM 053*, *VITF3L*, *EVM139*, *EVM141*, and *KBTB1* genes were amplified from the genome of the ECTV-Moscow strain and cloned into the pcDNA3.1-HA vector. Mutations were obtained by site-directed mutagenesis (D0206-QuickMutation Kit, Beyotime) from full-length HA-PAPL plasmids. The recombinant plasmids were transfected into *E.coli* DH-5α competent cells (9057, Takara, Beijing, China). Single clones were selected for PCR identification, and the plasmids were extracted from the positive clone of *E. coli* using the Endo-Free Plasmid Maxi kit (D6926, Omega Bio-Tek, Norcross, GA, USA). The plasmids were transfected into BSR-T7 cells using Lipofectamine 2000 (11668019, Invitrogen, Waltham, CA, USA) as per the manufacturer’s instructions. The primer sequences used for constructing the plasmids can be found in Appendix A.

### 4.6. Viral Genome DNA Copy Number Test

For absolute quantification, the viral genomic DNA was extracted from both the infected cell culture supernatants and whole-cell lysates. The *EVM003* gene was engineered into T-vector to create the standard plasmids, which were then used to establish a standard curve. We used qPCR to measure the EVM003 copy number, and we calculated the viral DNA copy number in the sample through standard curve analysis.

### 4.7. Immunoprecipitation (IP)/Mass Spectrometry (MS)

RAW-264.7 cells were infected with ECTV at an MOI of 1 for 12 and 24 h. Following cell lysis, supernatants were subjected to immunoprecipitation (IP) as per the manufacturer’s instructions (IP & Co-IP Kit, 635721, Takara, Beijing, China). This kit contains specially designed Protein A columns using a novel Capturem technology for the simple and rapid capture of antibody-bound target protein complexes. Briefly, incubate 20 μL GBP1-5 monoclonal antibody with 800 μL cell lysate (5 × 10^6^ cells) at 4 °C for 1 h. Equilibrate the spin column with 100 μL Lysis/Equilibration Buffer. Load the pre-incubated sample onto the equilibrated spin column and centrifuge at 1000 g at room temperature for 1 min. Repeat twice. Add 100 μL Wash Buffer and centrifuge. Add 4 µL neutralization buffer (1M Tris, pH8) to the collection tube. Then add 40 µL Elution Buffer to the spin column. Centrifuge and vortex the eluate. The elution was analyzed using mass spectrometry (MS) by Sangon Biotech (Shanghai, China). The mass spectrometry data analysis has been submitted to iProX (www.iprox.org, Proteome Xchange ID: PXD044061) (accessed on 25 July 2023).

### 4.8. Co-Immunoprecipitation and Immunoblotting Analysis

Co-immunoprecipitation (co-IP) assays were performed by co-transfecting plasmids into HEK-293T cells using the calcium chloride transfection method. HEK-293T cells were plated in a 10 cm dish. Immunoprecipitation experiments were then performed following the instructions provided in the Dynabeads Co-immunoprecipitation Kit (14321D, Novex, USA). Basically, the antibody and magnetic beads were incubated at 37 °C for 16–24 h in a rotator. Cell lysate was then added and incubated at 4 °C for 30 min. The eluted magnetic beads were then resuspended in 60 μL EB and incubated for 5 min. Next, the centrifuge tube was placed on the magnetic separator for 1 min to allow the magnetic beads to be fully absorbed. The elution was collected for SDS-PAGE and immunoblotting. In brief, proteins were transferred to PVDF membranes using a semi-dry transfer method. The primary antibody was incubated with the membranes at 4 °C overnight. After being washed three times with TBST, the membrane was incubated with a horseradish peroxidase-conjugated secondary antibody for 1 h at room temperature. Protein bands were visualized with enhanced chemiluminescence reaction buffer (ECL) (34078, Thermo Fisher Scientific, Waltham, MA, USA) and imaged on a high-resolution image acquisition system (Bio-Rad, Hercules, CA, USA). The grayscale values of the immunoblotting band were measured using Image-Pro Plus 6.0 software.

### 4.9. GST Pull-Down

The *GBP2* gene was amplified from IFN-γ-stimulated RAW264.7 cells and subcloned into the pGEX-4T-1 expression vector. The primer sequences used were GST-GBP2-F-TCCCCCGGGGATGGCCTCAGAGATCCACATG (SmaI),GST-GBP2-R-ACGCGTCGACCTCAGAGTATAGTGCACTTCCCAG (Sal I). Rosetta competent cells were transformed with GST-GBP2 recombinant plasmid; positive colonies were induced with 0.5 mM IPTG overnight at 16 °C. The bacterial pellet was resuspended in PBS and dispersed by sonication, then centrifuged at 12,000 rpm for 10 min to obtain the supernatant. The successfully expressed proteins were stored at −80 °C for future use. Glutathione-agarose beads (1 mL) (17075601, GE Healthcare, New York, NY, USA) were washed with PBS. Subsequently, 2 mL of the GST-GBP2 lysate supernatant was added to the beads and incubated at 4 °C for 1 h with gentle vortex. The beads were then centrifuged at 500 g for 5 min, resulting in the binding of GST-GBP2 to the resin. The resin was washed three times with pre-chilled PBS, resuspended in pre-chilled PBS, and mixed with 20 μL solution of renatured His-PAPL protein. This mixture was incubated at 4 °C for 4–8 h. Simultaneously, purified GST protein was used as a negative control. After centrifugation at 500 g and washing five times with PBS, the beads were resuspended in 20 μL PBS. Then, 20 μL of 2 × SDS loading buffer was added, and the mixture was boiled for 10 min. The precipitated components underwent immunoblotting analysis using mouse anti-His (1:1000) (AE003, ABclonal, Wuhan, China) and mouse anti-GST PAb (1:2000) (A02030, Abbkine, Wuhan, China).

### 4.10. Immunofluorescence

Confocal imaging was performed on BSR-T7 cells plated on confocal glass-bottomed dishes. The plasmids were transfected into the cells with lipofectamine 2000. The cells were then fixed with 4% paraformaldehyde for 30 min and permeabilized with 0.2% Triton X-100 for 15 min, followed by blocking with 5% BSA for 1 h. The cells were incubated with anti-HA (1:400) and anti-Flag (1:400) antibodies for 1 h. Subsequently, they were stained with Alexa Fluor 647 goat anti-mouse (405322, BioLegend, San Diego, CA, USA) and Alexa Fluor 488 donkey anti-rabbit (406416, BioLegend) secondary antibodies. The samples were then embedded in DAPI and observed under a confocal microscope. The colocalization coefficients were analyzed using Image-Pro Plus 6.0 software. The essence of colocalization of fluorescence analysis is to analyze the spatial overlap of proteins labeled with different fluorescence (with independent emission wavelengths) to determine if the two proteins are in the same area, meaning they are on the same pixel. In this analysis, images obtained by merging the red channel (representing Flag-GBP2 or the mutant plasmids) and the green channel (representing HA-PAPL) are used. In brief, use Image-Pro Plus to open a colocalized image and click on ‘Measure’ and select ‘Co-Localization’. Since it involves the red and green channels, choose ‘Create color co-localization’ and ‘Red-Green’ in the pop-up window. Then, click ‘Forward’ to obtain a scatter plot formed by the red and green channels. Next, select ‘Calculate co-localization coefficients’ and click ‘Analyze’ to obtain various parameters. These parameters are used for regression analysis based on the scatter plot, resulting in the calculation of the overlap coefficient.

### 4.11. Statistical Analysis

The data in this paper are representative of three experiments and have been analyzed with the statistical methods described in the figure legends. All data are expressed as means ± SDs. Student’s *t* test, two-way ANOVA, or one-way ANOVA were used in the statistical analysis, and significant differences are indicated in figures using the following symbols: * *p* < 0.1; ** *p* < 0.01; *** *p* < 0.001; **** *p* < 0.0001; ‘ns’ is used to indicate no significance.

## 5. Conclusions

This study highlights the interplay between PAPL and GBP2. While ectopic expression of HA-PAPL alone does not significantly increase ECTV replication, it can reduce the expression of Flag-GBP2, thereby counteracting its antiviral activity. The antagonism is reduced when key active sites of PAPL are mutated. Knocking out PAPL in ECTV leads to increase GBP2 expression, and it also results in a detrimental effect on viral replication. Knocking down GBP2 cannot reverse this damage. These findings highlight the essential role of PAPL as a candidate viral factor for ECTV replication. Overall, the study not only enhances our understanding of poxvirus interactions with host defenses, but also identifies potential targets for antiviral interventions.

## Figures and Tables

**Figure 1 ijms-24-15750-f001:**
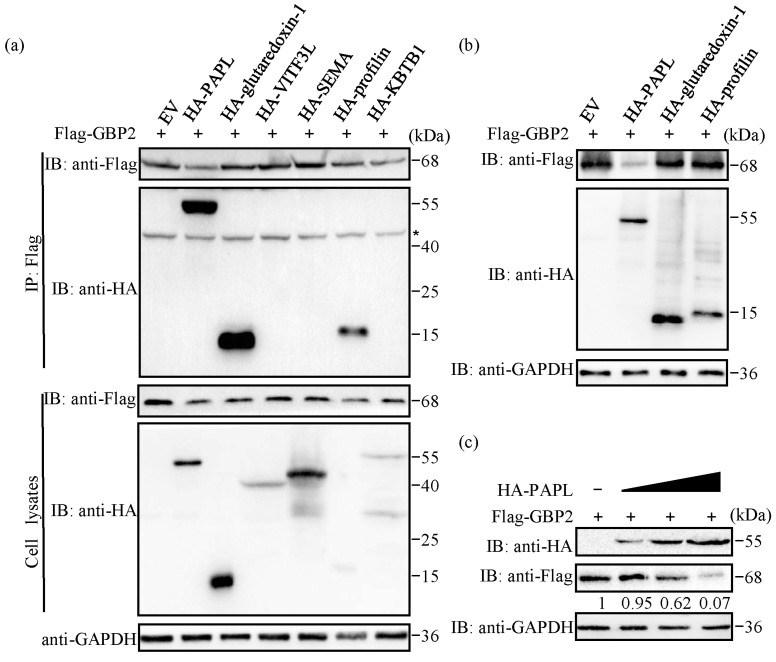
Identification of viral proteins that potentially interact with GBP2. (**a**) Flag-GBP2 and six HA-tagged viral proteins were co-transfected into HEK-293T cells, respectively, followed by co-IP using anti-Flag monoclonal antibody (mAb). GAPDH was used as internal reference. The nonspecific bands (asterisk) may be the endogenous IgG heavy chains. The pcDNA 3.1-HA empty vector (EV) were used as negative control. (**b**) HEK-293T cells in 6-well plates were co-transfected with 1.5 μg Flag-GBP2 and HA-EV, HA-PAPL, HA-glutaredoxin-1, or HA-profilin of equal quality, respectively. Samples were collected for Western blotting analysis. (**c**) HEK-293T cells were co-transfected with increasing doses of HA-tagged PAPL (0.5, 1.0, and 1.5 μg/mL) or the empty vector (1.5 μg/mL) along with 1.5 μg Flag-GBP2. The 0.5 and 1.0 μg/mL groups were supplemented with the empty vector to reach a final concentration of 1.5 μg/mL. After 24 h, samples were collected for Western blotting to determine protein level. The gray value was normalized to the first band (set as 1).

**Figure 2 ijms-24-15750-f002:**
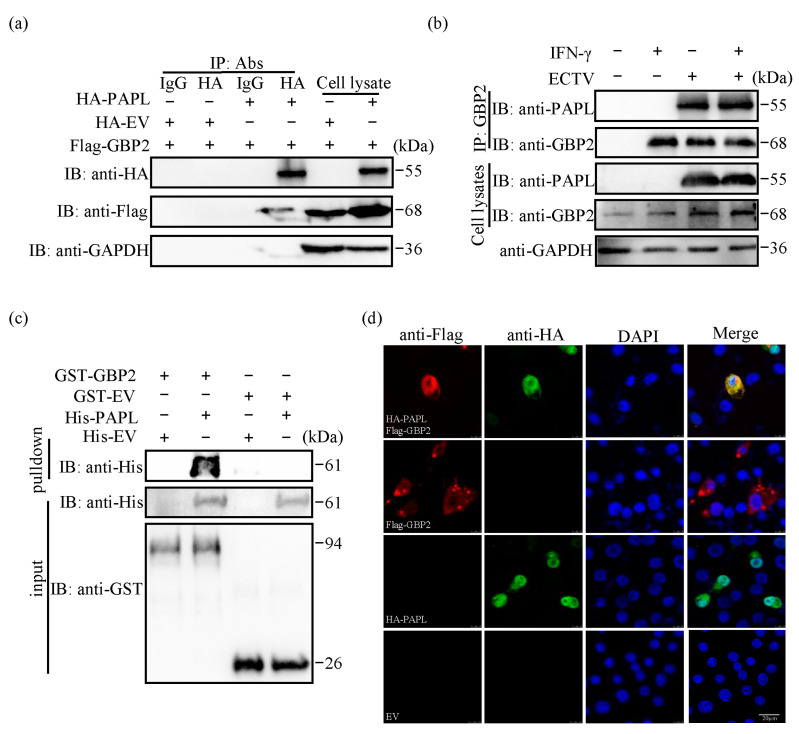
ECTV PAPL protein interacts with GBP2. (**a**) The interaction between HA-PAPL and Flag-GBP2 was analyzed by Co-IP with anti-HA PAb. HEK-293T cells were co-transfected with Flag-GBP2 (10 μg) and HA-PAPL (6 μg) expression plasmids or HA empty vector (6 μg). Samples were collected after 24 h and co-IP experiments were performed. (**b**) Endogenous identification of the interaction between GBP2 and PAPL in RAW264.7 cells. Cells were pretreated with or without IFN-γ (1000 U/mL) for 12 h, then infected or mock-infected with ECTV (MOI = 1). Co-IP assay was subjected using the anti-GBP2 antibody. The presence of PAPL and GBP2 proteins was analyzed through immunoblotting (IB) using anti-PAPL and anti-GBP2 antibodies. (**c**) GST-Pulldown. Purification of prokaryotic expressed GST-EV and GST-GBP2 with glutathione Sepharose resin. The resin was subsequently incubated with His-tagged PAPL, and the eluted proteins were analyzed by Western blotting using anti-His antibodies. (**d**) Colocalization of Flag-GBP2 with HA-PAPL. Flag-GBP2 (1.5 μg/mL) and HA-PAPL (1 μg/mL) were transfected alone or co-transfected into BSR-T7 cells for immunofluorescence analysis under a confocal microscope. The colocalization coefficients were analyzed using Image-Pro Plus 6.0 software. Scale bar: 20 µm.

**Figure 3 ijms-24-15750-f003:**
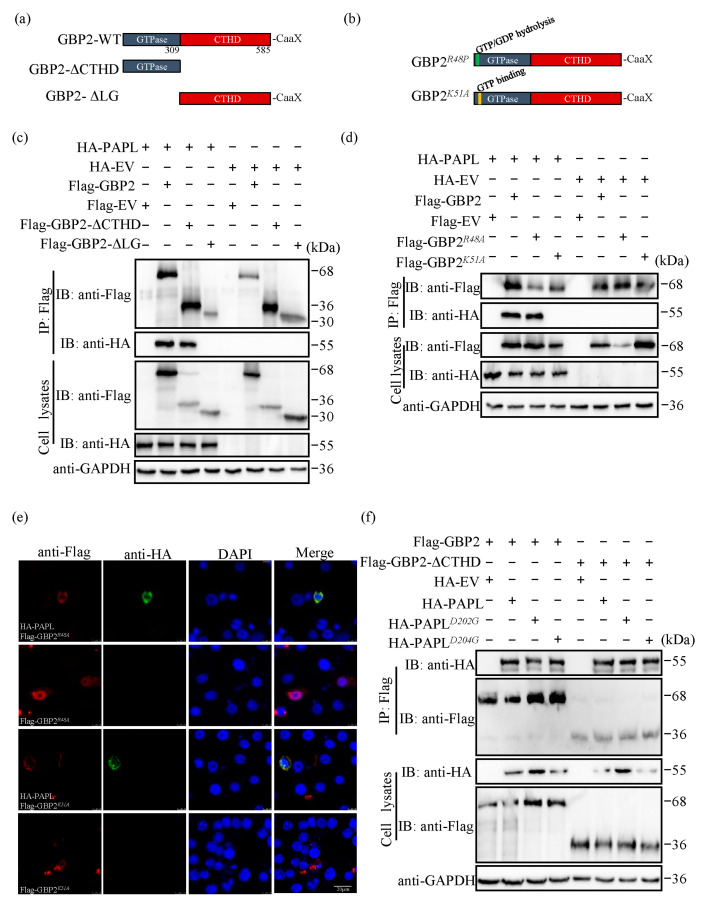
The K51 residue of GBP2 is essential for its interaction with PAPL. (**a**) Schematic presentation of full-length and truncated domain of GBP2. (**b**) Schematic presentation of the two mutants of GBP2. (**c**,**d**) HEK-293T cells were co-transfected with HA-PAPL and either full length or various truncated mutant Flag-tagged GBP2 expression plasmids. Empty vectors (EV) were used as controls. Co-IP was performed using anti-Flag mAb with the cell lysates. (**e**) Colocalization of Flag-GBP2*^R48A^* or Flag-GBP2*^K51A^* with HA-PAPL. Plasmids were transfected alone or co-transfected into BSR-T7 cells for immunofluorescence analysis. The colocalization coefficients were analyzed using Image-Pro Plus 6.0 software. Scale bar: 20 µm. (**f**) HEK-293T cells were co-transfected with either HA-PAPL or its mutant constructs HA-PAPL*^D202G^*/HA-PAPL*^D204G^*, along with full length Flag-GBP2 or truncated Flag-GBP2-ΔCTHD. The resulting cell lysates were used to perform co-IP with anti-Flag mAb.

**Figure 4 ijms-24-15750-f004:**
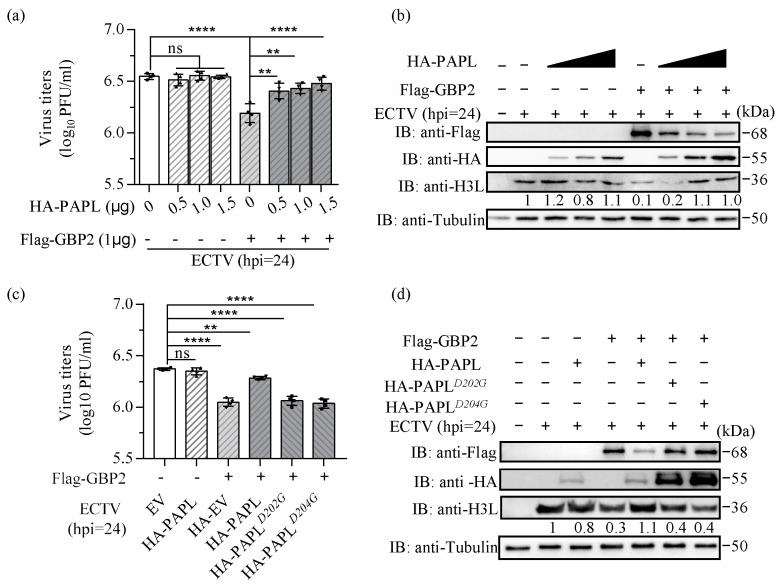
PAPL antagonizes the antiviral activity of GBP2. (**a**,**b**) BSR-T7 cells were co-transfected with increasing doses of HA-PAPL (0.5, 1.0, and 1.5 μg/mL) and Flag-GBP2 (1.0 μg/mL) or an pCDNA3.1-Flag empty vector (1.0 μg/mL). The pCDNA3.1-HA empty vector was supplemented to equalize the total plasmid concentration (2.5 μg/mL). The cells were then infected with ECTV (MOI = 1) for 24 h and analyzed using a plaque assay (**a**) and immunoblotting (**b**). (**c**,**d**) BSR-T7 cells were co-transfected with either HA-PAPL, HA-PAPL*^D202G^*, or HA-PAPL*^D204G^* and Flag-GBP2 for 16 h and then infected with ECTV (MOI = 1) for 24 h. The cells were then harvested separately for virus titer assay or protein level detection. ** *p* < 0.01; **** *p* < 0.0001; ‘ns’ is used to indicate no significance.

**Figure 5 ijms-24-15750-f005:**
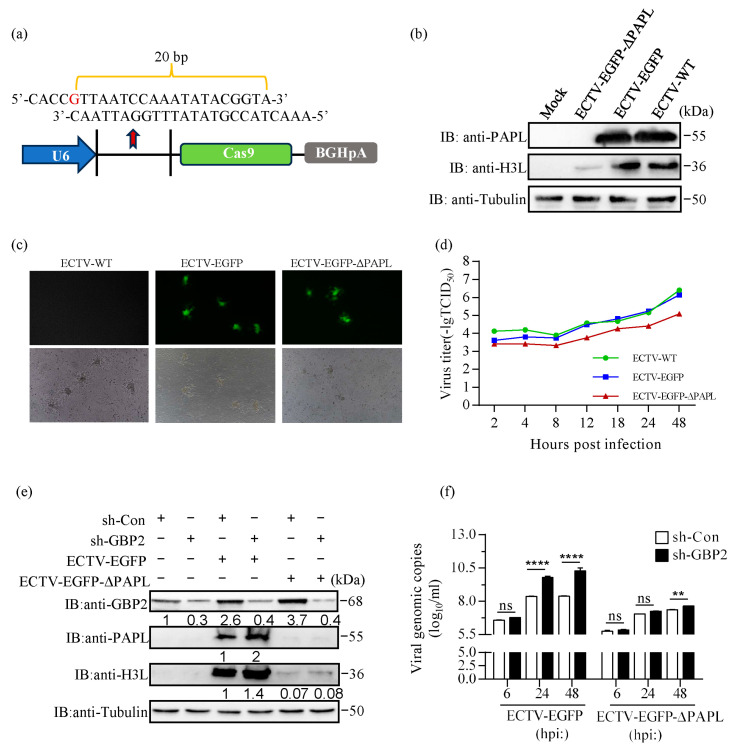
Deletion of the PAPL gene led to a reduction in virus replication in vitro. (**a**) Schematic diagram of vector construction using the CRISPR-Cas9 system to target and disable the PAPL gene. (**b**) The expression of H3L and PAPL proteins in ECTV-EGFP-ΔPAPL, ECTV-EGFP parent virus, and ECTV-WT. BSR-T7 cells were infected with different viruses for 24 h; uninfected cells served as a mock control. Western blotting analysis was performed on cell lysates to detect the indicated proteins. (**c**) Viral plaque formation in the CV-1 cells was visualized by fluorescence microscopy. Different viruses (MOI = 0.01) were used to infect CV-1 cells grown in 6-well plates. After 2 h of infection, the cells were covered with carboxymethylcellulose and incubated at 37 °C for 48 h. (**d**) The viral titer was measured using the standard TCID_50_ assay to generate a multi-step growth curve. (**e**,**f**) The screened sh-GBP2 and sh-Con cell lines were infected with ECTV-EGFP-ΔPAPL or ECTV-EGFP (MOI = 0.1), respectively. The viral genome DNA and protein were extracted separately for verification. ** p < 0.01; **** p < 0.0001; ‘ns’ is used to indicate no significance.

**Table 1 ijms-24-15750-t001:** Potential interaction viral proteins of GBP2 obtained by IP/MS.

Gene	Description	Mw(kD)	%Cov (95)	Number of Peptides (>95%)
** *PAPL* **	**Poly(A) polymerase large subunit (PAPL)**	**55.5**	**4.38**	**3**
*EVM053*	Glutaredoxin-1	12.3	8.33	2
*VIT3L*	Intermediate transcription factor 3 large subunit (VIT3L)	44.6	10.98	3
*EVM139*	Semaphorin-like protein (SEMA)	45.4	5.76	3
*EVM141*	Profilin	15.2	7.46	1
*KBTB1*	Kelch repeat and BTB domain-containing protein 1(KBTB1)	64.5	4.44	2

The protein of interest is in bold.

## Data Availability

All data are reported in the manuscript.

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
