# Peer review of "The Viral Protein Poly(A) Polymerase Catalytic Subunit Interacts with Guanylate-Binding Proteins 2 to Antagonize the Antiviral Ability of Targeting Ectromelia Virus"

_ijms, 2023, doi:10.3390/ijms242115750_

Round 1

Reviewer 1 Report

Comments and Suggestions for Authors

The current manuscript, by Zhenzhen et al, utilizes the Ectromelia virus (ECTV) as a virus model to study orthopoxvirus infections, a rising concern due to the recent spread of the monkeypox virus. A previous study conducted by the authors reported an up-regulation of GBP proteins induced by ECTV infection. GBP proteins regulate cell growth and resistance to pathogens. Namely, GBP2 was reported to cause an inhibitory effect on ECTV infection.  This manuscript is a follow-up study aimed to investigate possible mechanisms in ECTV that enable this virus to counteract the GBP2 anti-virulence effect. Immunoprecipitation/mass spectrometry analysis was carried out to identify ECTV proteins that physically interact with GBP2 in mouse and human cell lines. Among the ECTV proteins identified as interactors of GBP2 was the PAPL protein, the catalytic subunit of a viral poly(A) polymerase. A battery of co-immunoprecipitation experiments along with immunofluorescence led to mapping the interaction sites into PAPL protein and suggested that the underlying mechanism implied a PAPL-induced proteasomal degradation of GBP2. Given the relevance of the study of viruses, especially  in a “post-pandemic” time, and the detailed mechanism proposed herein to alleviate anti-viral immunological responses, I consider that the manuscript might be of interest to the scientific community, particularly, the virologist and immunologist. Nonetheless, the manuscript harbors several major and minor observations that must be addressed prior to considering it suitable for publication. 

Major comments:

The hypothesis that the authors tested is that ECTV proteins might interact with GBP2 protein attenuating ECTV virulence. After several experiments, the conclusion was that the ECTV’s PAPL protein physically interacts with GBP2 and such interaction exerts an attenuation of GBP2 anti-ECTV effect. The underlying mechanism is that PAPL-GBP2 interaction induces GBP2 protein degradation. This is further analyzed through the implementation of several autophagy and proteasomal inhibitors, concluding that the degradation is carried out via proteasomal-manner. I consider imperative to perform a qualitative and/or quantitative detection of ubiquitinated GBP2 in response to PAPL expression since polyubiquitination is the main signaling mechanism for proteasomal protein processing. One way to perform this experiment is to carry out a co-immunoprecipitation of GBP2 and PAPL and measure the levels of ubiquitinated GBP2. Otherwise, the mechanism that controls GBP2 protein levels by its interaction with PAPL remains elusive. 

Minor comments:

1.- In section 2.1, there is a change in the use of cell lines which is not explained. The authors performed the initial IP analysis with RAW264.7 cells and for subsequent IP experiments the HEK-293T cell line was utilized. If there is an experimental or conceptual reasoning for the cell line change it should be addressed in this section. Moreover, the HEK-293T cells are not mentioned in this section, therefore it might be assumed that all the IP experiments are performed with RAW264-7 cells. Only in the figure capture of Figure 1 the use of HEK293T cells is noted. Additionally, the authors use BSR-T7 cells with no explanation about the change in the use of cell lines. 

2.- Lines 113 through 116 are read as follows: “In order to investigate the interaction between GBP2 and the six viral proteins, we co-transfected them with GBP2 individually and conducted co-immunoprecipitation (co-IP) experiments using the Flag monoclonal antibody. The findings revealed that PAPL, glutaredoxin-1, and profilin exhibited interaction with GBP2”. Here the authors jump from the depiction of the experimental approach to the conclusion with no description of the results at all. I would expect a description of the differential presence of six viral proteins in the IP vs. the cell lysate fractions. Additionally, the phrase “In order to investigate the interaction between GBP2 and the six viral proteins, we co-transfected them with GBP2 individually” is misleading, since it does not mention which cells were transfected and the fact that the authors are referring to protein might be interpreted that the protein was transfected, instead of plasmids used for transient transfection of genes of interest. This would be clarified if a nomenclature such as pGBP2-FLAG is used to refer to the use of plasmids. Also, it is necessary to differentiate the gene name with italic letters (GBP2) and normal letters (GBP2) when referring to protein.

 3.- In Lines 116-121, it is not clear in which fractions (IP, or cell lysate) the decrease of GBP2 and its dose-dependent reduction is detected.

4.- In Fig 1 it is necessary to mention the use of GAPDH as a reference protein for the IP analyses.

5.- In Fig 1, there is a ≈40 kDa protein that appears in every analyzed IP fraction. Do the authors have an idea about the identity of that band? Regardless of the identification, it should not be omitted in the results section. Also, it is necessary to define the label EV within the figure capture.

6.- Although in figure capture of Fig.2c the authors specify the used recombinant protein (“prokaryotic expressed” GST fusion proteins), that is not mentioned in the results. Also, it is necessary to define the IB label and mention the details of the overexpression and purification of the recombinant proteins in the Materials and Methods section.

7.- In Figure 2d and Lines 147-149 where the colocalization of HA-PAPL and Flag-GBP2 proteins is reported to be mapped in the cytoplasm, given the high DAPI staining it is quite notorious the marginal transfection efficiency of BSR-T7 cells. The colocalization and expression of HA-PAPL and Flag-GBP2 individually are detected only in one cell. Therefore, it is necessary to report the efficiency of transformation that can be quantified based on the fluorescent-positive cells. 

8.- In lines 171-177 it is mentioned the use of point mutant of Flag-GBP2 (R48A and K51A) and PAPL-HA (D204E and D202E) plasmids to map the sites for BCP2-PAPL interaction. Nonetheless, there is no biochemical or structural explanation for the implementation of such mutations in the study. Actually, the Asp to Glu change is a conservative mutation which not necessarily disturb drastically PALP’s function. Although GBP2’s D202, D204, and K51 residues are mentioned in the introduction section, it is important to justify the rationale of these mutations in the results section.

9.- The Y-axis segmentation in figure 4a and 4c jumps from 6 to 6.0. Normally, the Y-axis segmentation is used when two or more bars with different scales are compared, which is not the case in this figure. Therefore, the Y-axis segmentation is not recommended. 

10.- Section 2.5 depicts the use of three knockout vectors designed specially targeting PAPL, with no information on the vector design. Actually, the authors just mention a frame-shift mutation near the active site of the PAPL gene. Moreover, there is no clear depiction of where the ECTV-EGFP-PALP and -ΔPALP virus strains are derived in this section. Only in the material and methods these virus strains are mentioned.  

11.- Lines 314-315: “Our study confirmed that PAPL can interact with GBP2 and strongly reduce its expression”. This manuscript’s hypothesis is that PAPL induces proteasomal-dependent GBP2 degradation. In such cases changes in the expression of GBP2 at transcript or protein are not measured, instead, the protein turnover seems to be affected by proteasomal proteolysis. Therefore, I recommend using the term protein levels instead of protein expression when referring to the effect that PAPL exerts on GBP2. And do so for the corresponding lines within the discussion and results sections.

12.- In lines 345-346 the authors state that “In order to understand how PAPL regulates the reduction of GBP2 expression, we used different protein degradation inhibitors, and confirmed that PAPL degrades GBP2 through the ubiquitin-proteasome pathway”. Here the authors should be more precise with the term 1) “protein degradation inhibitors” and the wording 2) “PAPL degrades GBP2”. Since they implemented three inhibitors; one of them is 3-MA, which is a P13K kinase that affects lipid turnover during autophagy. Also, they used MG132, a well-known proteasome inhibitor, and NH4Cl which is reported to inhibit proteolysis through depression of cathepsin B. Therefore, experimentally, the authors tested two different mechanisms of protein level regulation in addition to the MG132-mediated proteasomal inhibition. On the other hand, the wording “PAPL degrades GBP2” is misleading. According to the results and hypothesis, PAPL does not degrade GBP2, it induces GBP2 degradation via proteasomal-manner instead.

13.- In the material and methods section, lines 443-424; there is no description of how the recombinant proteins used for the immunization of rabbits were produced. 

Author Response

Dear Editors and Reviewer:

We feel great thanks for your professional review work on our article. As you are concerned, there are several problems that need to be addressed. According to your nice suggestions, we have made extensive corrections to our previous draft.Please see the attachment.

  sincerely     zhenzhen GAO  

Reviewer 2 Report

Comments and Suggestions for Authors

In the manuscript "The viral protein poly(A) polymerase catalytic subunit interacts with guanylate-binding proteins 2 to antagonize the antiviral ability of targeting ectromelia virus" by Gao et.al., the authors report the role of viral Poly (A) polymerase (PAP) in attenuating the anti-viral ability of host Guanylate Binding Proteins 2 (GBP2). Using a variety of immunoprecipitation and co expression experiments, the authors show a direct interaction of PAP to GBP2 and the ability of the viral PAP to target GBP2 for degradation thereby attenuating its antiviral ability. However, GBP2 knock down experiments and expression of PAP only showed modest increase in viral replication. The manuscript is overall well written, and the data well presented. However, several concerns in experiments leading to the claims made in the manuscript need clarification. Specific comments are listed below. 

1. In the results describing confocal imaging experiments (Figs 2d and 3e), the authors report a high degree of co-localization (coefficient values reported) between GBP2 and PAPL. How did the authors arrive at this result/conclusion? There is no description of how this was calculated in the methods section. What is the resolution of the confocal microscope? Lines 147-149 and 178-179: What do the authors mean when they state "HA-PAPL and Flag-GBP2 were both located in the cytoplasm and exhibited a high degree of colocalization"? Do they mean that both proteins are in the cytoplasm or both proteins are co-localized, as in interacting with each other. These are two different things. If they are both present in the cytoplasm, simply because that is where they are localized does not necessarily mean they are interacting with each other. The authors should clarify this point. If they are just present in the cytoplasm, it certainly is supportive of interaction based on the IP experiments but is not evidence in of itself of interaction. A better way to show interaction in vivo would be to add a LOV domain with an NLS to PAPL and show that GBP2 shuttles into the nucleus upon activation with blue light. Since the authors have access to a confocal microscope this will be a powerful way to show interaction in vivo. 

2. What fraction of PAPL is bound by GBP2 in vivo? This will be insightful in understanding the functional mechanism of GBP2 mediated inhibition of viral replication.  The authors can easily estimate this by comparing the bound and unbound fractions from their IP experiments.

3. The ability of GBP2 to inhibit virus titer in the absence of PAPL is significant, but modest (Fig 4a). If GBP2 was the prime attenuator of viral replication, it would be expected to have a more significant impact in inhibiting virus replication. This might be suggestive of additional proteins necessary for this inhibitory effect by GBP2, that is now limiting in the presence of enhanced levels of GBP2. Surprisingly, expression of PAPL alone does not alter the viral titer in BSR-T7 cells (Figure 4a). The absence of any effect of PAPL when expressed in the absence of GBP1 is intriguing. One would expect some enhancement of viral titer in the presence of PAPBL. How do the authors explain this?

4. In RAW264.7 cells, even upon knockdown of GBP2 (Figure 5e), there is no significant loss of GBP2 when PAPL is expressed. In fact, in GBP2-Con cells, there is enhanced expression of GBP2 when PAPL is co-expressed, 1 vs 2.6 fold and there is no significant change in GBP2 knockdown cells even in the presence of PAPL. This is contrary to the argument that GBP2 is marked for proteosomal degradation by PAPL claimed from the data presented in Figure How do the authors reconcile these opposing data?

5. In the data presented in Figure 5f, Control cells appear to have a significant loss of viral genome copies when transfected with PAPL deletion compared to full length PAPL. How do the authors reconcile this. Overall, both control and GBP2 deletion cells show consistently lower levels of viral replication relative to cells transfected with full length PAPL. It is not clear how this experiment was done. Is the virus replication levels reported in this experiment that of the PAPL virus itself? If so, what is the rationale for this, as opposed to using another virus as a reporter. 

6. A schematic representation of the GBP2 truncations used in experiments presented in Figure 3 will be immensely helpful.

Author Response

Dear Editors and Reviewers:

We sincerely appreciate the valuable comments. As you are concerned, there are several problems that need to be addressed. We apologize for any confusion regarding the reviewer's question, and hope that the explanation below adequately addresses the reviewer's query.

  sincerely         zhenzhen GAO

Round 2

Reviewer 1 Report

Comments and Suggestions for Authors

After the minor and major concerns that the authors addressed for the manuscript, I consider it suitable for publication.

Author Response

Dear reviewer:
        Thank you very much for the encouraging coments.  We sincerely appreciate the valuable feedback that we have used to improve the quality of our manuscript. Best wishes!

         sincerely 
          Jing

Reviewer 2 Report

Comments and Suggestions for Authors

The authors have failed to adequately address queries 1, 2 and 3.

It appears the authors are confused about query #1. The authors have added a line to the methods section to indicate teh correlation coefficient was calculated by using ImagePro Plus Software, but do not give any description of how the analysis was performed. In addition, the scale bar indicated in the figure is 20 uM, significantly large to provide any sense of correlation of co-localization of the two protein in a cell. Interestingly, in the cell these two proteins appear to be localized in the cytoplasm (see merged cell in top panel of Figure 3e) and very little PAPL in the nucleus. One implication being, PAPL is not in the nucleus to perform the polyadenylation function. The authors fail to comment on this important result. This also raises another question, do all the cells where both plasmids were transfected show the same kind of results presented in Figure 3e. The results presented in shows only 1 cell in the field of view where there is localization of both proteins in the cytoplasm! 

Query #2. In line with the comment above, the response to this query does not address the level of interaction of PAPL with GBP2. Specifically, what fraction of PAPL is bound by GBP2 in cells? A simple experiment to compare the total levels of GBP2 in the cell to how much of it is bound to PAPL when immunoprecipitated will be suggestive of the functional outcome of this interaction. Instead, the authors appear to have confused this with the truncations of GBP2 and mutants of PAPL they have addressed in the study.

Query #3. It is not clear where the changes mentioned by the authors have been made. 

In the response letter, the authors do not make clear what their response is to each query, and how and where the text or experiment was modified to address the queries. 

Author Response

Dear reviewer:

We are very grateful for your comments regarding our manuscript. All your suggestions are very important to us, both for composing the manuscript and our further research. We have studied comments carefully and have made corrections which we hope meet with approval. Please see the attachment.

Sincerely

Jing

Round 3

Reviewer 2 Report

Comments and Suggestions for Authors

I read the revised manuscript and find the authors have revised the manuscript satisfactorily. The manuscript is suitable for acceptance if other reviewer(s) and or the editor do not have any objections and are on board.